# Characteristics of Bacterial Community and Function in Paddy Soil Profile around Antimony Mine and Its Response to Antimony and Arsenic Contamination

**DOI:** 10.3390/ijerph16244883

**Published:** 2019-12-04

**Authors:** Bocong Huang, Jian Long, Hongkai Liao, Lingfei Liu, Juan Li, Jumei Zhang, Yirong Li, Xian Wang, Rui Yang

**Affiliations:** 1Guizhou Provincial Key Laboratory for Information System of Mountainous Areas and Protection of Ecological Environment, Guizhou Normal University, Guiyang 550001, China; huangbocongg@163.com (B.H.); liaohongkaii@163.com (H.L.); liulfsuns@126.com (L.L.); mrzhangjumei@163.com (J.Z.); liyirongaa@163.com (Y.L.); wx1996ll@163.com (X.W.); yangruia1995@163.com (R.Y.); 2School of Geography and Environmental Science, Guizhou Normal University, Guiyang 550001, China; lijuan_113@126.com

**Keywords:** antimony and arsenic contamination, paddy soil bacterial function, co-occurrence network analysis, LEfSe analysis, PLS-PM analysis

## Abstract

Research of bacterial communities and metabolism potential of paddy soils contaminated by antimony (Sb) and arsenic (As) are vital to acquire understanding for their bioremediation. Here, the relative abundance of Sb and As metabolism genes, the diversity and composition of the bacterial community, and the influences of geochemical properties and the bacterial community and metabolism potential have been researched by Tax4Fun2 prediction and high-throughput sequencing. LEfSe (linear discriminant analysis effect size) analysis shown different taxa were enriched in dissimilar soil layers. RDA (Redundancy analysis) and relative importance analysis indicated the main properties including total sulfur (TS), total organic carbon (TOC), pH, and the bioavailable fractions of Sb and As affects the bacterial community, which Sbrec, Astot, and Asrec had greater impact on the bacterial taxonomic community. For example, Asrec, Astot, and Sbrec had a positive correlation with Chloroflexi and Rokubacteria, but negatively correlated with Proteobacteria and Actinobacteria. Obtaining metabolic function genes by using the tax prediction method. RDA, relative importance analysis, and co-occurrence network analysis showed the geochemical properties and bacterial community affected Sb and As related bacterial functions. The partial least squares path model (PLS-PM) analysis indicated Sb and As contamination fractions had negative effects on ecological function, bacterial community structure had positive influences on ecological function, and the direct effects of geochemical properties on ecological function was greater than community structure. The direct impact of As contamination fractions on bacterial community structure was greater than Sb, while the direct impact of Sb contamination fractions on bacterial function was more remarkable than As. Obviously, this study provides a scientific basis for the potential of biochemical remediation of Sb and As contamination in paddy soils profile.

## 1. Introduction

As a non-essential toxic trace metal element in organisms, antimony (Sb) has been listed as a priority pollutant by the United States Environmental Protection Agency (EPA) and the European Union [1]. Due to the combined action of surface runoff, biological processes and atmosphere as well as the lack of an effective management, Sb from mining and smelting process is released into the surrounding supergene environment. Moreover, with Sb enrichment, migration, and transformation, which cause a considerable contamination in the mining and smelting areas as well as surrounding environments. Antimony (Sb) and arsenic (As) have similar physical and chemical properties and have related propertied, therefore, As contamination often accompanies the periphery of the Sb ore. China is the world’s largest producer of antimony, with Guizhou accounting for 10.2% of the total production [2]. The Sb content in environmental media around the Dushan Antimony mine (in Guizhou Province, southwest China) is much higher than the background value. For example, the concentration of Sb in the soil can reach 100,000 mg·kg^−1^, and that in sediments can reach 16,017 mg·kg^−1^ around Banpo Antimony Mine [3]. The high concentration of antimony contamination may endanger crops and even human health around antimony mining and smelting areas [3,4,5,6].

Microorganisms are critical factors in trace elements migration, transformation, methylation, and remediation because of their bioavailability of trace elements. It was reported that Sb widely existed in the lithosphere and was liable to form symbiotic sulfides with As [7], and then form complex pollution of Sb and As in the surrounding environmental media of a mining area [8]. Many reports have been made on the biochemical cycling process and biochemical mechanism of As in different environmental media [9,10,11,12,13]. In recent years, the effects of Sb contamination on microbial diversity and community structure of surrounding environmental media have aroused people’s wide attention. Xiao et al. (2016) studied the characteristics of microbial communities and their responses to Sb and As in Dushan Sb tailings contaminated watershed [14]. Sun et al. (2017) studied the effects of different concentrations of Sb contaminated soil around Banpo Antimony Mine on microbial community composition and diversity [15]. Previous studies have found that migration and transformation of Sb and As can influence the distribution of microorganisms in environmental media, causing significant differences in microbial community structure [16,17,18]. In addition, the diversity and community structure of microorganisms in paddy soils were significantly affected by the different distribution of Sb forms. The activities of these microorganisms greatly affected the transformation of Sb and As in paddy soils. 

The relative abundance, diversity, and activity of genes related to Sb and As transformation often determine the migration and transformation of Sb and As. Reductive genes (arrA and arsC), which affect the reduction of As (V) and mediate respiratory and detoxification pathways, oxidase genes (aioA) and small subunits of oxidase (aoxA and aoxB) promote the oxidation of As (III), and transferase genes (arsM, arsA, arsB, and ACR3) mediate migration, transformation, and methylation pathways [9,10]. Although the mechanism of As migration and transformation has been well studied in microbial culture [19,20], little is known about the abundance, diversity, and activity of genes related to Sb and As transformation in paddy soils, the enrichment of related genes in the migration and transformation of Sb and As, and whether the related genes are affected by Sb and As contamination fractions.

In this study, we selected the paddy soils adjacent to Banpo Antimony Mine and a smelter in Dushan County, southwest China. In order to understand the effect of Sb and As contamination fractions as well as other geochemical properties on the structure and function of bacterial community in paddy soil profile, our hypothesis was that the characteristics of the bacterial community and function in different soil layers and its responses to Sb and As contamination were different. To test the hypothesis that Sb and As contamination fractions in paddy soil, we measured the bacterial communities and functions by using a 16S rRNA (The DNA sequence corresponding to rRNA encoded in bacteria exists in the genome of all bacteria) sequencing technique. The aims of this study are: (1) to understand the distribution of Sb and As contamination fractions as well as other geochemical properties in paddy soil profile; (2) to clarify the distribution characteristics of bacterial communities and functions in paddy soil profile; and (3) to determine the potential relationship between Sb and As contamination fractions as well as other geochemical properties and soil bacterial communities and functions.

## 2. Material and Methods

### 2.1. Site Description

Sampling sites were located in the paddy soils around Banpo Antimony Mine and Xiaohe Antimony Mine Smelter in Dushan County, Guizhou Province, Southwest China. Banpo Antimony Mine has been in operation since the last century, and Xiaohe Antimony Mine Smelter has been smelting since 1970. Therefore, the accumulation of Sb and As has been present in the paddy soils around them. Five soil profiles with a width of 1 m and a depth of 1 m were collected in the paddy soils around the antimony mining area and the smelting area. Two soil samples of the profiles were collected from the Banpo antimony mining area, whereas three soil samples were collected from the smelting area. Moreover, the profile from the surface to the bottom was three layers: SL1 (0–30 cm), SL2 (30–65 cm), and SL3 (65–100 cm). Paddy soil was greatly affected by human activities, and according to the classification of soil taxonomy [21], the paddy soil can be divided into either Aqualfs or Aquepts. SL1, SL2, and SL3 were the submergenic horizon, percogenic horizon, and waterloggogenic horizon, respectively.

### 2.2. Geochemical Properties Analysis

All paddy soil samples were pre-treated after being naturally air-dried. Plant roots, leaves, and gravels form the paddy soil were ground and passed through 2 mm sieve. Then these soil samples were further thoroughly ground and passed through a 200-mesh sieve. Total sulfur, total organic carbon, and the carbon–nitrogen ratio were determined by an elemental analyzer (vario MACRO cube; Elementar, Hanau, Germany); and pH was used to determine the glass electrode method (water–soil ratio 2.5:1). Total Fe was determined as described previously [22,23]. Briefly, 1 g soil sample were mixed with 10 mL 1.0 mol·L^−1^ of HCl for 30 min with shaking, and allowed to equilibrate for 4 h. The supernatant was then collected by centrifugation at 3500 rpm for 10 min. Finally, the total Fe concentration was determined by 1,10-phenanthroline spectrophotometry at 510 nm.

### 2.3. Antimony and Arsenic Contamination Fractions Analysis

The contamination fractions of Sb and As were determined by BCR continuous extraction method [24,25]. Briefly, the acid exchangeable fraction was extracted using 40 mL 0.11 mol·L^−1^ of CH_3_COOH. Next, the reducible fraction was extracted using 40 mL 0.5 mol·L^−1^ of NH_2_OH·HCl. The oxidizable fraction was then extracted by 20 mL 1.0 mol·L^−1^ of CH_3_COONH_4_ after oxidization using 10 mL 8.8 mol·L^−1^ of H_2_O_2_.The supernatant was injected into a polyethylene container and stored in a refrigerator at 4 °C for testing. Then HG-AFS (hydride generation-atomic fluorescence spectrometry, AFS-933, Jitian, Beijing) was used to determine Sb and As contamination fractions. Duplicate samples (repeated three times for each sample), reagent blanks, and standard reference materials (Certified Reference Materials for the Chemical Composition for Soil GSS 28, Chinese Academy of Geological Science, Beijing, China) were included during the procedure to ensure the analytical accuracy. The percentage difference of the duplicate samples was under ±5%, and the metal recoveries of Sb and As in the standard reference materials were 92% and 95%, respectively.

### 2.4. DNA Extraction and PCR Amplification

Microbial DNA was extracted from soil samples using the E.Z.N.A.^®^ soil DNA Kit (Omega Bio-tek, Norcross, GA, USA) according to the manufacturer’s protocols. The final DNA concentration and purification were determined by NanoDrop 2000 UV-vis spectrophotometer (Thermo Scientific, Wilmington, DE, USA), and DNA quality was checked by 1% agarose gel electrophoresis. The V3–V4 hypervariable regions of the bacterial 16S rRNA gene were amplified with primers 338 F (5’-ACTCCTACGGGAGGCAGCAG-3’) and 806 R (5’-GGACTACHVGGGTWTCTAAT-3’) by the thermocycler PCR system (GeneAmp 9700, ABI, Applied Biosystems, Foster City, CA, USA). The PCR reactions were conducted using the following program: 3 min of denaturation at 95 °C, 27 cycles of 30 s at 95 °C, 30 s for annealing at 55 °C, and 45 s for elongation at 72 °C, and a final extension at 72 °C for 10 min. PCR reactions were performed in triplicate 20 μL mixture containing 4 μL of 5 × FastPfu Buffer, 2 μL of 2.5 mM dNTPs, 0.8 μL of each primer (5 μM), 0.4 μL of FastPfu polymerase, and 10 ng of template DNA. 

The resulting PCR products were extracted from a 2% agarose gel and further purified using the AxyPrep DNA Gel Extraction Kit (Axygen Biosciences, Union City, CA, USA) and quantified using QuantiFluor™-ST (Promega, Madison, WI, USA) according to the manufacturer’s protocol. Purified amplicons were pooled in equimolar and paired-end sequenced (2 × 300) on an Illumina MiSeq platform (Illumina, San Diego, CA, USA) according to the standard protocols by Majorbio Bio-Pharm Technology Co. Ltd. (Shanghai, China).

### 2.5. Data Analysis

Soil bacteria 16S rRNA data processing: The original sequencing sequence was spliced by QIIME2 [26] and DADA2 [27] software to remove the barcode and primer. According to the similarity level of 0.97, the naive Bayesian algorithm was used to train the feature classifier based on Silva database (SSU132 version, Max Planck Institute for Marine Microbiology and Jacobs University, Bremen, Germany. Species annotation files were obtained by further training on behalf of sequence and feature classifier. Sequence variation files (amplicon sequence variant table, ASV) were matched with annotation files to get the classification level of species.

The environmental factors and bacterial community structure were analyzed by PCoA (principle coordination analysis) and RDA using the ape package [28], vegan package [29], psych package [30], and reshape2 package [31] in R language. The relaimpo package was used to calculate the relative importance of environmental factors to species [32]. The igraph package was used to analyze the co-occurrence network of bacterial community structure, and Gephi 0.9.2 software (company, city, country) was used to draw the network map [33]. PLS-PM package (company, city, country) was used to calculate the direct or indirect relationship between soil bacterial function, soil physical, and chemical properties [34], and bacterial community structure. Ggplot2 package (company, city, country) draws bacterial diversity map and bubble map [35]. Tax4Fun2 package (company, city, country) was used to predict the ecological functions of representative documents and feature tables [36]. All the above analyses were performed in R 3.4.2 (AT&T Bell Laboratory, America). Using Python 2.7 software (National Institute of Mathematics and Computer Science, city, Netherlands), LEfSe analysis of bacteria was carried out.

## 3. Results

### 3.1. Geochemical Properties

The major geochemical properties of the paddy soil samples collected from Dushan antimony mine are listed in Figure 1. The concentrations of total Sb (Sbtot) and total As (Astot) in 15 samples from different layers of paddy soils were higher, ranging from 4.47 to 124.91 mg·kg^−1^ in Sbtot, and from 30.84 to 92.75 mg·kg^−1^ in Astot, respectively. In the bioaccessible fractions of Sb, the exchangeable fraction (Sbaec) and the oxidizable fraction (Sboxz) increased with the soil depth increase, and the reducible fraction (Sbrec) remained unchanged with an increase of soil depth. In the bioaccessible fractions of As, the concentration of fractions remained the same as the soil layers increased. Other geochemical properties of the various layers of the paddy soil profile were compared (Figure 1), the pH value varied from 5.9 to 8.0. With the increase of soil depth, the pH value changed from weak acidity to weak alkalinity. The contents of total sulfur (TS), total nitrogen (TN), total organic carbon (TOC), and total Fe (Fetot) decreased with the increase of soil depth.

### 3.2. Bacterial Community Diversity and Structure Analysis

A total of 20 phyla, 66 classes, 153 orders, 243 families, and 325 genera were detected in five profiles. Chloroflexi, Proteobacteria, and Actinobacteria were the dominant bacteria in profiles of paddy soils, accounting for 51.58% to 83.33% of the bacterial community. Chloroflexi increased with depth, while Proteobacteria and Actinobacteria decreased with profiles (Figure 2A). Principal coordinate analysis of dominant bacteria in the top 10 bacterial communities (Figure 2B) showed that the first and second principal axes explained 39.51% and 15.02% respectively. In addition, the bacterial community composition of SL2 and SL3 were higher than that of SL1, and the dominant bacteria of SL1 was Actinobacteria, the dominant bacteria of SL2 was Proteobacteria, and the dominant bacteria of SL3 was Chloroflexi. The community structure of SL1 and SL2, SL1, and SL3 were different, but the difference between SL2 and SL3 was small, and the similarity was large. From the diversity index distribution chart (Figure 2C), ace, Chao1, Shannon, and Simpson indices in SL1, SL2, and SL3 show the same regularity, the largest in SL1, the smallest in SL3, and SL1 > SL2 > SL3.

The diversity of taxa and distribution characteristics of species in paddy soils, LEfSe analysis of SL1, SL2, and SL3 bacteria from phylum to genus was carried out (Figure 3). Linear discriminant analysis (LDA) only displayed distinctively indicated species with a significant threshold greater than 3.0. The results showed that 23 differential indicators were detected in three soil layers, and 12, 2, and 9 were detected in SL1, SL2, and SL3, respectively. Among them, Rokubacteria and NC10, Proteobacteria, and Actinobacteria were the indicators with the most significant difference among soil bacterial taxa of SL1, SL2, and SL3.

### 3.3. The Relationship between Geochemical Properties and Bacterial Communities

The bacterial community of different soil layers in the paddy soil profile was quite different, which may be the result of the interaction of Sb and As contamination fractions and other geochemical properties. RDA analysis was carried out on geochemical properties at the phylum level (Figure 4A), to explore the coupling relationship between bacterial community and geochemical properties. The first and second principal axes explained 51.50% and 18.26% variance, respectively. TS, TOC, and Sbtot had a positive correlation with Proteobacteria and Actinobacteria, but negatively correlated with Chloroflexi and Rokubacteria. Asrec, Astot, and Sbrec had an active correlation with Chloroflexi and Rokubacteria, but negatively correlated with Proteobacteria and Actinobacteria. Overall, the contamination fractions of Sb and As, TS, TOC, and pH had an obvious effect on the bacterial community structure. In addition, quantitatively analysis of geochemical properties on the phylum level as shown in Figure 4B. The results showed that Astot had a more evident influence on the dominant bacterial phylum such as GAL15, Proteobacteria, and Verrucomicrobia. Sbrec and Asrec had more overt impact on Firmicutes, Chloroflexi, and Verrucomicrobia. TS and TOC had more notable effect on Rokubacteria, and Acidobacteria.

### 3.4. The Relationship between Geochemical Properties and Bacterial Function

Tax4Fun2 was used to predict 7272 kinds of functional enzymes, including carbon, nitrogen, sulfur, and iron. 11 functional enzymes related to Sb and As including arsR, arsC2, ACR3, arsC1, AS3MT, arsA, arsB, arsH, arsC, aoxA, and aoxB. The relative abundances of aoxA and aoxB in different soil layers are lower, while the relative abundances of arsR, arsC2, and ACR3 in different soil layers were higher. Furthermore, the relative abundance of arsR, arsC2, and ACR3 between SL1 and SL2 was higher than that of SL3 (Figure 5A). According to PCoA analysis (Figure 5B), the composition of functional enzymes in SL1 and SL2 was slightly different. On the contrary, there was significantly different in SL1 and SL3, SL2, and SL3. The relationship between geochemical properties and Sb and As related functional enzymes was further shown by RDA analysis (Figure 5C) and relative importance analysis (Figure 5D). RDA analysis showed that Sbtot, TS, TOC, and Fetot had a positive correlation with arsC2. Asrec. Similarly, pH and Sbrec were positively correlated with arsR and ACR3. The results of relative importance shown that TS, TOC, and Fetot had more considerable influence on arsR. Similarly, Fetot, Sboxz, and Asoxz had a more significant influence on arsC2, and Sbtot, Asrec, and Sbrec had a more significant effect on ACR3.

### 3.5. Relationship with Community Structure and Bacterial Function

RDA analysis and relative importance analysis showed that bacterial community structure and functional enzymes were correlated with Sb and As contamination fractions and other geochemical properties. The co-occurrence network was used to visually show the correlation between species and functional enzymes at the classification level of soil bacteria genus in SL1, SL2, and SL3 (Figure 6A–C). The modularization coefficients of SL1, SL2, and SL3 are 1.692, 1.651, and 1.376 respectively; the average clustering coefficients are 0.373, 0.351, and 0.358; and the average path lengths were 3.364, 4.001, and 3.641, respectively. According to the function of nodes, SL1, SL2, and SL3 had four, three and two functional modules, respectively. These functional modules included most of the nodes in each soil layer. All nodes of bacterial community structure and functional enzymes were divided into six modules, of which the four main functional modules contained the major of nodes (Figure 6D). Among them, module I contained ACR3, arsH, arsC, AS3MT, arsB, and five genera; module II contained arsA, arsC1, and five genera; module III contained aoxA and six genera; and module IV contained arsR, arsC2, and three genera.

### 3.6. Determining the Relationship between Geochemical Properties, Bacterial Community, and Function

Microbial communities and functional redundant species at different taxonomic levels can ensure the stability of ecosystem processes in response to environmental disturbances [37]. For understanding the difference of bacterial functional redundancy in different soil layers of paddy soil, the functional redundancy index (FRI) in Tax4Fun2 was introduced to compare soil functional enzymes in different soil layers of paddy soil (Figure 7A–C). Higher FRI indicated that specific functional enzymes existed in various species of the bacterial community, while lower FRI indicated that the functional enzymes existed only in a few specific species, and none FRI indicated that there are no such functional enzymes at all. In order to display the FRI values of different soil layers better, the absolute values were obtained after logarithmic conversion. The larger the values (except infinite), the higher the difference of FRI values between the two layers [36]. The results showed that the FRI of SL1 was higher than that of SL2, the FRI of SL1 was higher than that of SL3, and the FRI of SL2 was higher than that of SL3.

Geochemical properties and bacterial community structure affected bacterial function in different soil layers of the paddy soil. Partial least square path model (PLS-PM) was established to explore the direct or indirect relationship between soil bacterial function, soil physicochemical properties, and bacterial community structure (Figure 7D). Soil geochemical properties were TS, TOC, Fetot, pH, and TN as well as Sb and As contamination fractions. Bacterial community structure refers to the top 10 dominant bacterial phyla in paddy soil, and soil bacterial function involved in Sb and As functions. The results showed that between Sb and As contamination fractions had a direct negative effect on soil bacterial function, and, however, other geochemical properties have a direct positive effect on it. The bacterial community structure had a direct positive effect on soil bacterial function, and its path coefficient was smaller than the environmental properties (0.483 < 1.1956). Sb and As contamination fractions and other geochemical properties indirectly affected soil bacterial function through bacterial community structure, and the path coefficients were 0.1946 and 0.6877, respectively. In addition, the interpretation rate of Sb and As contamination fractions and other geochemical properties on bacterial function was 0.863, and the statistical value of the goodness of fit (GoF) statistical evaluation model was an indicator for predicting overall performance, and its value was 0.5698.

## 4. Discussion

### 4.1. Distribution Characteristics of Geochemical Properties in Paddy Soil Profiles

Even though there were relatively more studies on antimony mining waste and surrounding soil, sediment, and water environmental media [3,38,39,40]. However, the distribution characteristics of Sb and As contamination fractions and other geochemical properties are not clear in the vertical profile of paddy soils. Continuous extraction was a standard method for estimating trace elements in soils and sediments [41,42,43,44], whereas Maec was considered to have a weak correlation with carbonates and exchangeable cations, usually associated with higher mobility [45]. Mrec was generally considered to be a reducible part associated with Fe/Mn oxygen/hydroxides, susceptible to redox conversion [46]; and Moxz was considered to be associated with organic compounds and sulfides, which was easy to transform under oxidation conditions. These contamination fractions were considered biologically available [47]. The potential and bioavailability of different fractions of contamination released into the environment were different [48]. In this study, Sbaec increased with the soil layer depth, which may indicate that the exchangeable cations decreased with the increase of the soil layer, and the passivation ability of trace elements in the soil layer decreased gradually, so the mobility of Sbaec decreased (lai, 2010). The pH value increased with the increase of soil depth, which might be due to the formation of hydroxides easily by Fe (II) and Fe (III) in soil under the oxidizing environment, which increased the pH value. In the acidic environment, the migration and transformation of Sb contamination fractions were relatively stable, while in weak alkaline condition, the Fe (OH)_2_ and Fe (OH)_3_ formed by soil had strong adsorption capacity for Sbrec, so that the content of Sbrec was still high in SL2 and SL3. The concentration of Sboxz increased with the depth of soil layer, which might be due to the accumulation and transformation of other contaminated fractions of Sb by microorganisms. With the increase of soil depth, the content of Sbtot decreased gradually, while the As contamination fractions remained basically unchanged with the increase of soil layer. Since the chemical properties of Sb and As were similar, it could be seen that the sources of Sb and As might be different in paddy soil profiles. Sb contamination might come from atmospheric deposition or artificial release, while As contamination might come from parent rock, which was similar to previous research results [49].

### 4.2. Effects of Sb and As Contamination Fractions on Bacterial Community

Trace elements contamination affected the community structure, richness, and diversity of bacteria in environmental media. High concentrations tended to reduce bacterial abundance and diversity [50,51], while bacteria were considered to be the most sensitive group to Sb and As contamination [52]. He et al. (2019) found that the content of Sb and As was less than 50 mg·kg^−1^ and had little effect on the composition of bacterial community [53]. Our results illustrated that bacterial diversity index decreased with soil depth increase, and there was no significant difference, which may be because the bacterial diversity in different soil layers was less affected when the concentrations of Sb and As were not up to 50 mg·kg^−1^. PCoA analysis showed that there were significant differences between SL1 and SL2, SL1 and SL3 in the bacterial community composition in different layers of paddy soil, and the similarities between SL2 and SL3 were higher. LEfSe analysis further suggested that different taxa were enriched in different soil layers, which reflected the characteristics of the soil layer to some extent, which was similar to previous research results [49]. The extractable fractions of Sb and As only accounted for a small fraction of Sbtot and Astot, which also significantly affected the bacterial community structure, indicating that low concentrations of Sb and As could still affect the activities of soil bacteria [54]. Previous reports have confirmed that a variety of microorganisms can better detoxify or metabolize under Sb and As contamination conditions [55,56]. In our study, the effects of Sbrec, Astot, and Asrec on soil bacterial community structure in different layers of paddy soil were more significant than those in other contamination fractions. RDA analysis showed that Sbrec, Astot, and Asrec were positively correlated with Chloroflex and negatively correlated with Proteobacteria. Relative importance analysis showed that Chloroflexi was strongly influenced by Sbrec, Astot, and Asrec, while Proteobacteria was mainly influenced by Astot. According to previous reports, Anaerolineaceae, as a specific anaerobic of Chloroflexi, existed in various environmental media contaminated by Sb and As [57,58], and Anaerolineaceae has also been reported to be widely distributed in contaminated paddy soil in China, Britain, and Bangladesh [59]. We found that Anaerolineaceae was the most abundant family of Chloroflexi and it decreased with the deepening of soil layer. 

### 4.3. Effects of Bacterial Community on Sb and As Related Function

There are some bacterial taxa with multiple functions in different ecosystems [60]. In this work, functional redundancy was at a high level in paddy soil layers, and the index was generally SL1 > SL2 > SL3, indicating that SL1 had more enzymes and higher abundance. However, the functional enzymes were less abundance in SL3, which might be the weakest in response to environmental changes such as Sb and As contamination. It was the redundancy of bacterial function that weakened the correlation between the bacterial community structure and function highlighting the role of soil environmental properties in the bacterial function.

Co-occurrence network analysis showed that genus1 of Anaerolineaceae was positively correlated with As-related genes such as arsC1, arsC2, arsA, and arsR, indicating that it might participate in the reduction and migration transformation process of Sb and As in the biochemical cycle, which further confirmed that Asrec had a tremendous positive impact on Chloroflexi. *Anaeromyxobacter* belongs to Proteobacteria, which was a typical As-reducing bacteria. It was found that As was detected mostly in contaminated soil. It has the function of metabolizing various electron receptors and can reduce As (V) [61]. *Anaeromyxobacter* mainly existed in SL3, and increased with the deepening of the soil layer. Through symbiotic network analysis, we found that *Anaeromyxobacter* (genus11) had a negative correlation with arsenate reductase (arsC, arsC1, and arsC2), which further confirmed that Astot has a negative impact on Proteobacteria. *Geobacter* [62], and *Sphingomonas* [63] were typical As metabolic bacteria. *Sphingomonas* had a positive response to As contamination fractions but had no response to Sb contamination fractions. It was further showed that Proteobacteria was apparently greatly affected by the As contamination fractions during the migration and transformation of Sb and As.

### 4.4. Effects of Geochemical Properties on Sb and As related Bacterial Function

The prediction of Tax4Fun2 indicated that microorganisms mediated the redox reaction of As in soil layers. Due to the similar chemical structure of Sb and As, soil microorganisms may migrate and transform Sb and As through similar metabolic pathways. The relative abundance of arsR, arsC2, and ACR3 in different soil layers was higher and decreased with the increase of soil layers. The reason was that the resistance genes to Sb contamination were weakened by the decreased concentration of Sb. It was reported that inactivation of arsR results in *Staphylococcus xylosus* plasmid pSX267 to Sb (III) and As (III) resistance decrease [64]. RDA analysis and relative importance analysis showed that arsR and arsC2 were strongly influenced by Sbrec, Sbaec, and Asrec, and were positively correlated, indicating that arsR and arsC2 existed in Sb and As-related reducing bacteria. Both Sb and As could induce arsenic-resistant operons containing arsR, and arsenic pump membrane protein (arsB). Therefore, the reduction of Sbrec might contribute to the enrichment of As-related reductase in this study, which was consistent with the results of Xiao et al. (2017), on the characteristics of bacterial communities structure in two different profiles contaminated by Sb and As [49].

Similarly, the oxidation of Sboxz may also contribute to the enrichment of As oxidase (aoxA and aoxB) or the presence of oxidase-like enzymes. Previous reports have found that *Agrobacterium tumefaciens* can oxidize Sb (III) [65], although aoxA and aoxB enzymes were not detected, suggesting that there might be other oxidation pathways for Sboxz. This further explained the relatively low abundance of aoxA and aoxB. RDA analysis and relative importance analysis demonstrated that arsB was strongly influenced by Sboxz and Asoxz, and positively correlated with Sboxz and Asoxz. The relationship might be due to the existence of arsB related enzymes in Sb and As oxidizing bacteria. Previous reports found that arsB [66], existed in Sb oxidizing bacteria *Comamonas*, while Meng et al. (2004), found that arsB could catalyze the transport of Sb (III) [67]. These results suggest that As related genes may also participate in the biogeochemical cycle of Sb. However, due to the lack of sequencing data related to Sb biological cycle in relevant databases, no biogeochemical cycle or resistance genes directly related to Sb was predicted.

Through the above RDA analysis and relative importance analysis, we found that both Sb and As contamination fractions and other geochemical properties affected the bacterial community structure and function. In order to further understand how bacterial function was affected by biological and abiotic factors in paddy soils, PLS-PM was used to describe the direct and indirect relationships between different variables. The results show that the direct impact of As contamination fractions on bacterial community structure was greater than that of Sb, which was consistent with previous study [49]. The direct impact of Sb contamination fractions on bacterial function was greater than that of As, because Sb has greater ability to shape bacterial community structure than As. While Wang et al. (2018) found that Sb was more critical than As in shaping bacterial community structure [68]. For bacterial function, bacterial community structure and soil environmental properties can be directly considered as internal and external factors. In this study, the effect of external factors on bacterial function was greater than that of internal factors, which was similar to the results of Zhou et al. (2019) on Soil and Bahia grass bacterial function [69]. Besides, although As related metabolism genes might participate in the geochemical cycling of sb, due to the limited representativeness of cultured isolates of sb metabolizing bacteria in the KEGG (Kyoto Encyclopedia of Genes and Genomes) database, the orthology genes of sb metabolism were not found. In addition, the knowledge of predicting the metabolic function of bacteria in paddy soil was preliminary and further study is needed.

## 5. Conclusions

In this paper, we quantified the effects of Sb and As contamination fractions and other geochemical properties on bacterial community and function in the vertical profile of paddy soils. Soil layer increased with decreasing Sb contamination fractions, while the As contamination fractions remained basically unchanged. The difference between Sb and As contamination fractions in different soil layers made different taxa of the bacterial community enrich in different soil layers. This enrichment might change with the vary of Sb and As contamination fractions. The main factors affecting the structure of bacterial community were TS, TOC, and pH as well as the contamination fractions of Sb and As, and the stability of soil bacteria. It was followed by SL1 > SL2 > SL3 when facing environmental changes such as Sb and As contamination in different soil layers. In addition, the direct influence of Sb and As contamination fractions on bacterial function was more significant than that of bacterial community structure. The direct impact of As contamination fractions on bacterial community structure was higher than Sb, while the direct impact of Sb on bacterial function was higher than As contamination fractions. The focus of the effects of Sb and As contamination on bacterial community and function provided knowledge for the geochemical process and biotransformation of Sb and As, which allows the development of strategies for the biochemical remediation of Sb and As contamination in the vertical profile of paddy soils.

## Figures and Tables

**Figure 1 ijerph-16-04883-f001:**
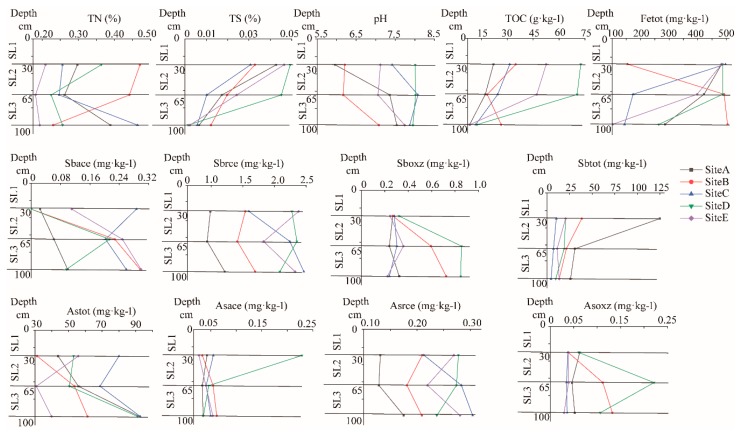
Vertical profiles of geochemical properties. Horizontal lines indicated three layers (SL1, SL2, and SL3) in paddy soil.

**Figure 2 ijerph-16-04883-f002:**
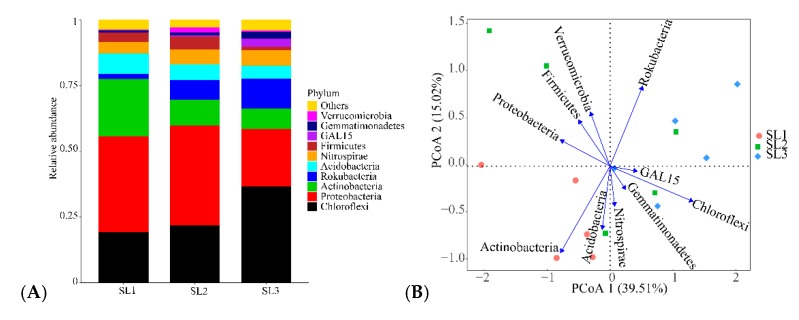
Relative abundance figures of soil bacteria at phylum level (**A**). PCoA analysis of soil microbial communities for different layers (**B**). Box plots indicating the alpha diversity indices for different layers in the paddy soil (**C**).

**Figure 3 ijerph-16-04883-f003:**
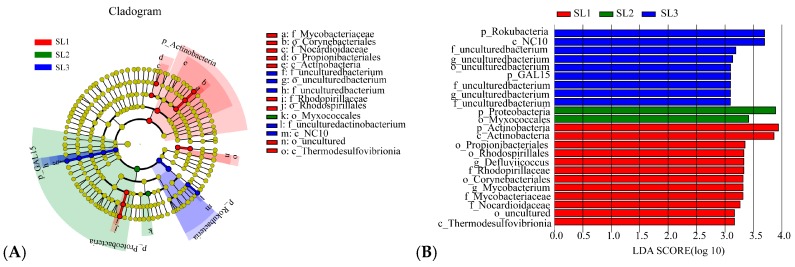
Linear discriminant analysis (LDA) effect size (LEfSe) shows the difference of soil bacteria in different layers of the paddy soil at different classification levels (**A**). The groups with significant differences are represented by red, green, and blue dots, and yellow dots represent groups with non-significant differences. From inside to outside, circles represent kingdoms, phyla, classes, orders, families, and genes. Only taxa with significant threshold LDA values >3.0 are shown in figures (**B**).

**Figure 4 ijerph-16-04883-f004:**
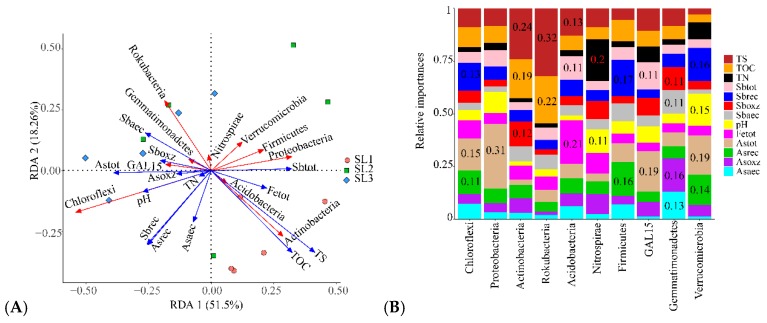
RDA analysis of bacterial phylum level and geochemical properties (**A**). Relative importance analysis of bacterial phylum level and geochemical properties (**B**).

**Figure 5 ijerph-16-04883-f005:**
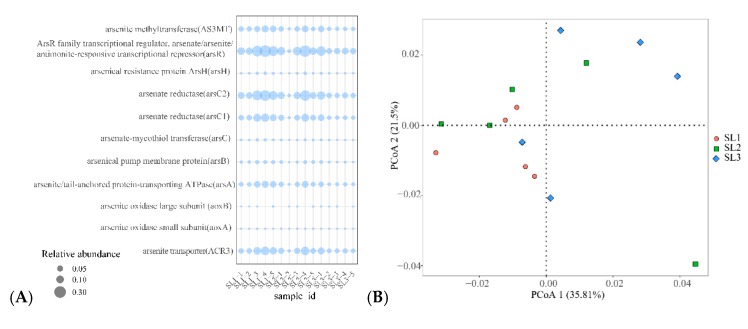
Tax4Fun2 based on 16S rRNA data predicts the relative abundance of Sb and As-related functional enzymes in different soil layers (**A**). PCoA analysis of Sb and As related functional enzymes (**B**). RDA analysis of geochemical properties and Sb and As related functional enzymes (**C**). Relative importance of geochemical properties and their related functional enzymes (**D**).

**Figure 6 ijerph-16-04883-f006:**
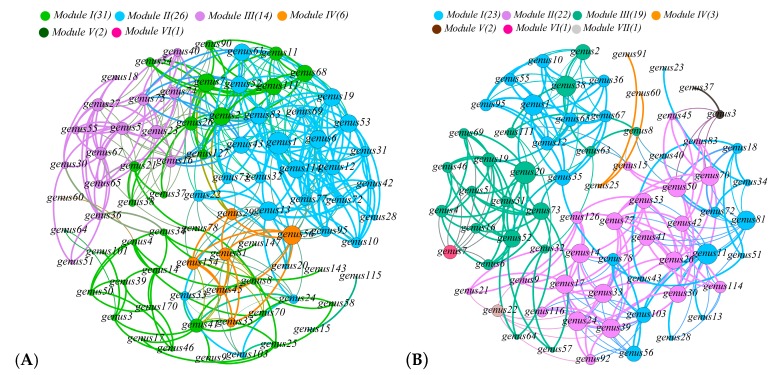
Co-occurrence network analysis showed the correlations among species of SL1, SL2, and SL3 bacterial genera at the taxonomic level in paddy soils (**A–C**). The correlations between functional enzymes and the genes taxonomic level (**D**). The edge only showed the Spearman correlation of strength (|r| > 0.6) and significance (*p* < 0.05); the size of each node was proportional to the number of connections (degree); the thickness of edge between two nodes was proportional to the value of Spearman correlation coefficient, ranging from |0.6| to |1|; and the symbiotic network was colored by modularization.

**Figure 7 ijerph-16-04883-f007:**
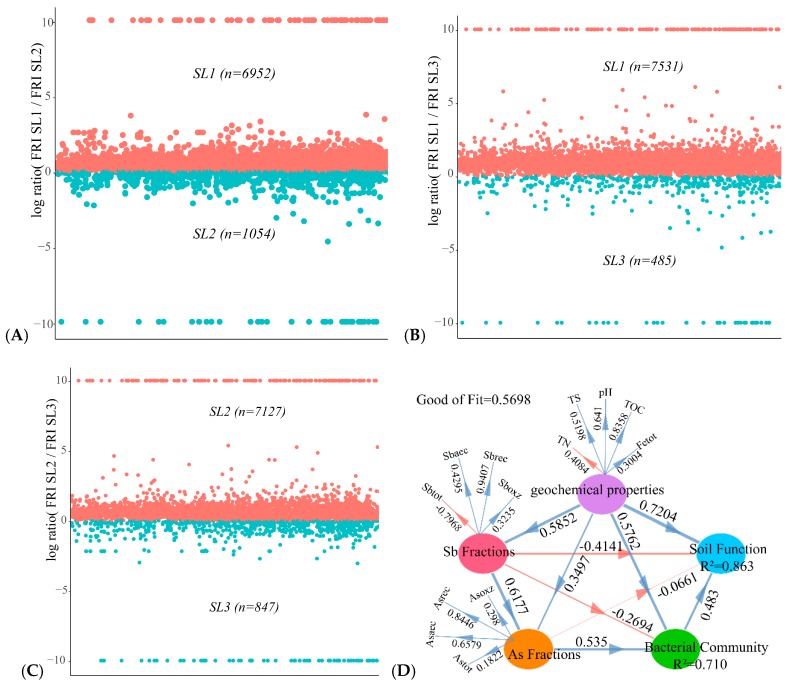
Comparing the bacterial functional redundancy index (FRI) of each soil layer, 97% similarity cut-off values were used for the prediction (10 and −10 replace positive and negative infinity, indicating that only one functional enzyme exists in different soil layers (**A**–**C**). The directed graph of the partial least squares path model (PLS-PM). The observed variables are represented by words and potential variables are represented by ellipses. After 1000 bootstraps, the path coefficients (between potential variables) and R^2^ (within ellipses) are calculated. The goodness of fit (GoF) statistical evaluation model is used to predict the overall performance (**D**).

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
