# Peer review of "Characteristics of Bacterial Community and Function in Paddy Soil Profile around Antimony Mine and Its Response to Antimony and Arsenic Contamination"

_ijerph, 2019, doi:10.3390/ijerph16244883_

Round 1

Reviewer 1 Report

Toxic trace elements pollution is one of the most important problems in the world. Especial Arsenic. This work determine the potential relationship between Sb and As contamination fractions and soil bacterial communities and functions. The background of this work is the most interesting. However, there are significant deficiencies in the manuscript.

There are still several typing and grammar errors and sentences with unclear meanings. Examples:

Line 53. First “As” should not an abbreviation, please check the context. When starting a sentence write the full name of an element.

Line80 what is “arrA and arsC”?

Line94 the “are” instead of “were”. (the objective section should be present tense)

Line 120 Add the quality control of Chemical analysis for Antimony and Arsenic

Line 172 some abbreviation words should note full name

Line 174 Five soil profiles of geochemical properties, should not show by Box-plot. I suggest should be improved. Such as vertical distribution map.

Introduction section should add some advances used method The “heavy metal” in the paper should be changed to trace element or heavy metal and As. What is the limitation of the study. I suggest should be shown at end of manuscript.

Author Response

Dear Reviewer,

We are truly grateful for your critical comments and thoughtful suggestions. Based on these comments and suggestions, we have made careful modifications on the downloaded the latest version manuscript from server and the revised version of the manuscript has been resubmitted to your journal. All changes made to the text are in red color. We hope that the revised manuscript will meet the journal’s standard. Below you will find our point-by-point responses to the comments/questions:

Responses to Reviewer:

Point 1: Line 53. First “As” should not an abbreviation, please check the context. When starting a sentence write the full name of an element.

Response 1: Thank you very much for your valuable comments and suggestions, according to your suggestion, we have modified the abbreviation.

Point 2:  Line80 what is “arrA and arsC”?

Response 2: Thank you very much for your valuable comments and suggestions. arrA and arsC are Reductive genes, this Manuscript has not been explained before, but has been modified. The specific expression in this paper is: Reductive genes (arrA and arsC), which affect the reduction of As (V), mediate respiratory and detoxification pathways

Point 3: Line94 the “are” instead of “were”. (the objective section should be present tense)

Response 3: Thank you very much for your valuable comments and suggestions. We have misunderstood the tense of this sentence before and have now revised it.

Point 4: Line 120 Add the quality control of Chemical analysis for Antimony and Arsenic

Response 4: Thank you very much for your valuable comments and suggestions. The quality control of Chemical analysis for Antimony and Arsenic is: Duplicate samples (repeat three times for each sample), reagent blanks and standard reference materials (Certified Reference Materials for the Chemical Composition for Soil GSS 28, Chinese Academy of Geological Science, Beijing, China) were included during the procedure to ensure the analytical accuracy. The percentage difference of the duplicate samples were under ±5%, and the metal recoveries of Sb and As in the standard reference materials were 92% and 95%, respectively.

Point 5: Line 172 some abbreviation words should note full name

Response 5: Thank you very much for your valuable comments and suggestions. Based on your suggestions, we have made corresponding modification. As below.

The major geochemical properties of the paddy soil samples collected from Dushan antimony mine are listed in Figure 1. The concentrations of total Sb (Sbtot) and total As (Astot) in 15 samples from different layers of paddy soils were higher, ranging from 4.47 mg·kg-1 to 124.91 mg·kg-1 in Sbtot, and from 30.84 mg·kg-1 to 92.75 mg·kg-1 in Astot, respectively. In the bioaccessible fractions of Sb, the exchangeable fraction (Sbaec) and the oxidizable fraction (Sboxz) increased with the soil depth increase, and the reducible fraction (Sbrec) remained unchanged with an increase of soil depth. In the bioaccessible fractions of As, the concentration of fractions remained the same as the soil layers increased. Other geochemical properties of the various layers of the paddy soil profile were compared (Figure 1), the pH value varied from 5.9 to 8.0. With the increase of soil depth, the pH value changed from weak acidity to weak alkalinity. The contents of total sulfur (TS), total nitrogen (TN), total organic carbon (TOC), and total Fe (Fetot) decreased with the increase of soil depth.

Point 6: Line 174 Five soil profiles of geochemical properties, should not show by Box-plot. I suggest should be improved. Such as vertical distribution map.

Response 6: Thank you very much for your valuable comments and suggestions. Based on your suggestions, we change the box-plot map to a vertical distribution map as below.

Figure 1. Vertical profiles of geochemical properties. Horizontal lines indicated three layers (SL1, SL2, and SL3) in paddy soil.

Point 7: Introduction section should add some advances used method the “heavy metal” in the paper should be changed to trace element or heavy metal and As. What is the limitation of the study. I suggest should be shown at end of manuscript.

Response 7: Thank you very much for your valuable comments and suggestions. Based on your suggestions, we have made corresponding modification. As for the limitation of the article research, we have added it at the end of the manuscript. As below.

In this study, although As related metabolism genes may participate in the geochemical cycling of sb, due to the limited representativeness of cultured isolates of sb metabolizing bacteria in KEGG database, the orthology genes of sb metabolism have not been found. In addition, the knowledge of predicting the metabolic function of bacteria in paddy soil is preliminary and further study is needed.

In addition, we have further updated the grammar and spelling errors in the manuscript. See the manuscript for details.

Finally, I would like to thank you for your valuable comments and suggestions in the review of the manuscript, and your valuable time in improving the quality of our manuscript.

Reviewer 2 Report

This is an article about the Characteristics of Bacterial Community in Paddy Soil around a Antimony Mine. The authors used high-throughput sequencing technology to analyze bacteria and the influences of Sb and As contamination fractions and other geochemical properties on bacterial community structure and metabolic potential.

In my opinión, although the paper is considered to be of scientific interest for the journal, it needs to undergo a major review for it to be aceptable.

I suggest to reduce the abstract making it more concise

In introduction, the objectives must be more specific and clearer;

In material and methods, what soil type are studied? (by Soil Taxonomy or FAO-UNESCO-ISSS).

What reference material was used to validate the analytical method?

I suggest to improve all the figures.

From pag 44 to 61 I suggest to introduce a separate point

In pag 171 use pH values with a unique decimal: i.e. 5.9 and 8.0

The sentence between line 337 and 376 is very long. Please introduce a separate point.

Finally, I suggest to include as reference: García R. and Jiménez Ballesta R. (2017) Mine tailings influencing soil contamination by potentially toxic elements. Environ Earth Sci 56-71. DOI 10.1007/s12665-016-6376-9

Author Response

Dear Reviewer,

We are truly grateful for your critical comments and thoughtful suggestions. Based on these comments and suggestions, we have made careful modifications on the downloaded the latest version manuscript from server and the revised version of the manuscript has been resubmitted to your journal. All changes made to the text are in red color. We hope that the revised manuscript will meet the journal’s standard. Below you will find our point-by-point responses to the comments/questions:

Responses to Reviewer:

Point 1: I suggest to reduce the abstract making it more concise

Response 1: Thank you very much for your valuable comments and suggestions. according to your suggestion, we have carefully checked and modified the abstract. The language and grammar are modified. At the same time, we reduced the number of words, increased the accuracy of the description and made it more concise. As below.

Abstract: The mining and smelting of antimony ore contaminate the surrounding paddy soil and bring a threat to food safety and human health. Research of bacterial communities and metabolism potential of paddy soils contaminated by antimony (Sb) and arsenic (As) are vital to acquire understanding for their bioremediation. Here, the relative abundance of Sb and As relate metabolism genes, the community diversity and composition of bacteria in paddy soils from antimony mining and smelting area of Dushan County, and the influences of between Sb and As bioavailable fractions as well as other geochemical properties and the bacterial community and metabolism potential have been researched by Tax4Fun2 functional prediction and high-throughput sequencing. LEfSe (Linear Discriminant Analysis Effect Size) analysis shown different taxa were enriched in dissimilar soil layers. RDA and relative importance analysis indicated the main properties including TS, TOC, pH and the bioavailable fractions of Sb and As affects the bacterial community, which Sbrec, Astot, and Asrec had greater impact on the bacterial taxonomic community. For example, Asrec, Astot, and Sbrec had a positive correlation with Chloroflexi and Rokubacteria, but negatively correlated with Proteobacteria and Actinobacteria. Obtaining metabolic function genes by using tax prediction method. RDA, relative importance analysis and co-occurrence network analysis shown geochemical properties and bacterial community affected Sb and As related bacterial functions. Also, the PLS-PM analysis indicated Sb and As contamination fractions had negative effects on ecological function, bacterial community structure had positive influences on ecological function, and the direct effects of geochemical properties on ecological function was greater than community structure. The direct impact of As contamination fractions on bacterial community structure was greater than Sb, while the direct impact of Sb contamination fractions on bacterial function was more remarkable than As. It is obviously that the potential of biochemical remediation of Sb and As contamination in paddy soils profile.

Point 2: In introduction, the objectives must be more specific and clearer

Response 2: Thank you very much for your valuable comments and suggestions. Based on your suggestion, we have carefully checked and modified the objectives of study. The language and grammar are modified. At the same time, increased the accuracy of the description and made it more concise. A detailed modification as follows.

In this study, we selected the paddy soils adjacent to Banpo Antimony Mine and a smelter in Dushan County, southwest China. In order to understand the effect of Sb and As contamination fractions as well as other geochemical properties on the structure and function of bacterial community in paddy soil profile, our hypothesis was that the characteristics of the bacterial community and function in different soil layers and its responses to Sb and As contamination were different. To test the hypothesis that Sb and As contamination fractions in paddy soil, we were measured the bacterial communities and functions by using 16S rRNA sequencing technique. The aims of this study are: 1) to understand the distribution of Sb and As contamination fractions as well as other geochemical properties in paddy soil profile; 2) to clarify the distribution characteristics of bacterial communities and functions in paddy soil profile; 3) to determine the potential relationship between Sb and As contamination fractions as well as other geochemical properties and soil bacterial communities and functions.

Point 3: In material and methods, what soil type are studied? (by Soil Taxonomy or FAO-UNESCO-ISSS).

Response 3: Thank you very much for your valuable comments and suggestions. Based on your suggestion, we added relevant content of soil taxonomy. A detailed modification as follows.

Paddy soil was greatly affected by human activities, according to the classification of soil taxonomy, the paddy soil can be divided into either alfisols or inceptisols. SL1, SL2 and SL3 were submergenic horizen, percogenic horizen and waterloggogenic horizen, respectively.

Point 4: What reference material was used to validate the analytical method?

Response 4: Thank you very much for your valuable comments and suggestions. Based on your suggestion, we have added reference materials as follows.

Duplicate samples (repeat three times for each sample), reagent blanks and standard reference materials (Certified Reference Materials for the Chemical Composition for Soil GSS 28, Chinese Academy of Geological Science, Beijing, China) were included during the procedure to ensure the analytical accuracy. The percentage difference of the duplicate samples were under ±5%, and the metal recoveries of Sb and As in the standard reference materials were 92% and 95%, respectively.

Point 5: I suggest to improve all the figures.

Response 5: Thank you very much for your valuable comments and suggestions. Based on your suggestions, we improved the figures.

Point 6: From pag 44 to 61 I suggest to introduce a separate point.

Response 6: Thank you very much for your valuable comments and suggestions. Based on your suggestion, we have carefully checked and modified the paragraph. At the same time, increased the accuracy of the description and made it more concise. A detailed modification as follows.

As a non-essential toxic trace metal element in organisms, antimony (Sb) has been listed as a priority pollutant by the United States Environmental Protection Agency (EPA) and the European Union. Due to the combined action of surface runoff, biology and atmosphere as well as the lack of an effective management, Sb from mining and smelting process is released into the surrounding supergene environment. Moreover, with Sb enrichment, migration, and transformation, which cause a considerable contamination in the mining and smelting areas as well as surrounding environments. Sb and arsenic (As) have similar physical and chemical properties and have related propertied, therefore, As contamination often accompany the periphery of Sb ore. China is the world's largest producer of antimony, with Guizhou accounting for 10.2% of total production. The Sb content in environmental media around Dushan Antimony mine (in Guizhou Province, southwest China) is much higher than the background value. For example, the concentration of Sb in the soil can reach 100,000 mg·kg-1, and that in sediments can reach 16017 mg·kg-1 around Banpo Antimony Mine. The high concentration of antimony contamination may endanger crops and even human health around antimony mining and smelting areas.

Point 7: In line 171 use pH values with a unique decimal: i.e. 5.9 and 8.0

Response 7: Thank you very much for your valuable comments and suggestions. Based on your suggestion, we have modified the contents.

Point 8: The sentence between line 337 and 376 is very long. Please introduce a separate point.

Response 8: Thank you very much for your valuable comments and suggestions. Based on your suggestion, this part has been modified. A detailed modification as follows.

4.2. Effects of Sb and As contamination fractions on bacterial community

Trace elements contamination affected the community structure, richness, and diversity of bacteria in environmental media. High concentrations tended to reduce bacterial abundance and diversity, while bacteria were considered to be the most sensitive group to Sb and As contamination. He et al. (2019), found that the content of Sb and As was less than 50 mg·kg-1 and had little effect on the composition of bacterial community. Our results illustrated that bacterial diversity index decreased with soil depth increase, and there was no significant difference may be because the bacterial diversity in different soil layers was less affected when the concentrations of Sb and As were not up to 50mg·kg-1. PCoA analysis showed that there were significant differences between SL1 and SL2, SL1 and SL3 in the bacterial community composition in different layers of paddy soil, and the similarities between SL2 and SL3 were higher. LEfSe analysis further suggested that different taxa are enriched in different soil layers, which reflected the characteristics of the soil layer to some extent, which is similar to previous research results. The extractable fractions of Sb and As, only account for a small fraction of Sbtot and Astot, that also significantly affect the bacterial community structure, indicating that low concentrations of Sb and As can still affect the activities of soil bacteria. Previous reports have confirmed that a variety of microorganisms can better detoxify or metabolize under Sb and As contamination conditions. In our study, the effects of Sbrec, Astot, and Asrec on soil bacterial community structure in different layers of paddy soil were more significant than those in other contamination fractions. RDA analysis showed that Sbrec, Astot, and Asrec were positively correlated with Chloroflex and negatively correlated with Proteobacteria. Relative importance analysis showed that Chloroflexi was strongly influenced by Sbrec, Astot, and Asrec, while Proteobacteria was mainly influenced by Astot. According to previous reports, Anaerolineaceae, as a specific anaerobic of Chloroflexi, existed in various environmental media contaminated by Sb and As, and Anaerolineaceae has also been reported to be widely distributed in contaminated paddy soil in China, Britain and Bangladesh. We found that Anaerolineaceae was the most abundant family of Chloroflexi and it decreased with the deepening of soil layer.

4.3. Effects of bacterial community on Sb and As related function

There are some bacterial taxa with multiple functions in different ecosystems. In this work, functional redundancy was at a high level in paddy soil layers, and index was generally SL1 > SL2 > SL3, indicating that SL1 had more enzymes and higher abundance. However, the functional enzymes were less abundance in SL3, which may be the weakest in response to environmental changes such as Sb and As contamination. It was the redundancy of bacterial function that weakened correlation between bacterial community structure and function highlighting the role of soil environmental properties in bacterial function.

Co-occurrence network analysis showed that genus1 of Anaerolineaceae was positively correlated with As-related genes such as arsC1, arsC2, arsA and arsR, indicating that it might participate in the reduction and migration transformation process of Sb and As in the biochemical cycle, which further confirmed that Asrec had a tremendous positive impact on Chloroflexi. Aneromyxobacteria belongs to Proteobacteria, which was a typical As-reducing bacteria. It was found that As is detected mostly in contaminated soil. It has the function of metabolizing various electron receptors and can reduce As (V). Anaeromyxobacter mainly existed in SL3, and increased with the deepening of soil layer. Through symbiotic network analysis, we found that Anaeromyxobacter (genus11) had a negative correlation with arsenate reductase (arsC, arsC1 and arsC2), which further confirmed that Astot has a negative impact on Proteobacteria. Geobacter, and Spengomonas were typical As metabolic bacteria. Sphingomonas has a positive response to As contamination fractions but has no response to Sb contamination fractions. It is further showed that Proteobacteria is apparent effected greatly affected by the As contamination fractions during the migration and transformation of Sb and As.

In addition, we have further updated the grammar and spelling errors in the manuscript. See the manuscript for details.

Finally, I would like to thank you for your valuable comments and suggestions in the review of the manuscript, and your valuable time in improving the quality of our manuscript.

Round 2

Reviewer 2 Report

Dear Authors

I suggest revising some sentences before publication:

Line 16-17. I suggest delette the first sentence

Line 19-24. The sentence:” Here, the relative abundance of Sb and As relate 20 metabolism genes, the community diversity and composition of bacteria in paddy soils from 21 antimony mining and smelting area of Dushan County, and the influences of between Sb and As 22 bioavailable fractions as well as other geochemical properties and the bacterial community and 23 metabolism potential have been researched by Tax4Fun2 functional prediction and high24 throughput sequencing.”  it's a very long phrase

Line 37-38. It´s obviously whtat?: It is obviously that the potential of 38 biochemical remediation of Sb and As contamination in paddy soils profile.

Line 45. Biology? Biological procesess?

Line 49. Sb and arsenic (As). I suggest to put Antimony (Sb) and arsenic

Line 105-108. “Paddy soil was greatly affected by human activities, according to the 106 classification of soil taxonomy[21], the paddy soil can be divided into either Alfisols or Inceptisols. 107 SL1, SL2 and SL3 were submergenic horizen, percogenic horizen and waterloggogenic horizen, 108 respectively” These are two phrases: 1 . Paddy soil was greatly affected by human activities, and 2. The rest…

In this same sentence: horizen or horizon?

If the soils are Alfisols or Inceptisols, logycally wil be Aqualfs or Aquepts. So I suggest to introduce this soil types at level or suborder.

Line 116: 10ml 1mol·L-1???

Line 223: “Overall, The contamination”, Overall, the contamination

Author Response

Manuscript ID: ijerph-642707

Title: Characteristics of bacterial community and function in paddy soil profile around antimony mine and its response to antimony and arsenic contamination

Dear Reviewer,

We are truly grateful for your critical comments and thoughtful suggestions. Based on these comments and suggestions, we have made careful modifications on the downloaded the latest version manuscript from server and the revised version of the manuscript has been resubmitted to your journal. All changes made to the text are in red color. We hope that the revised manuscript will meet the journal’s standard. Below you will find our point-by-point responses to the comments/questions:

Responses to Reviewer:

Point 1: Line 16-17. I suggest delete the first sentence

Response 1: Thank you very much for your valuable comments and suggestions. According to your suggestion, this sentence has been deleted.

Point 2: Line 19-24. The sentence:” Here, the relative abundance of Sb and As relate 20 metabolism genes, the community diversity and composition of bacteria in paddy soils from 21 antimony mining and smelting area of Dushan County, and the influences of between Sb and As 22 bioavailable fractions as well as other geochemical properties and the bacterial community and 23 metabolism potential have been researched by Tax4Fun2 functional prediction and high24 throughput sequencing.”  it's a very long phrase

Response 2: Thank you very much for your valuable comments and suggestions. Based on your suggestion, we have carefully checked and modified this sentence. A detailed modification as follows.

Here, the relative abundance of Sb and As metabolism genes, the diversity and composition of bacterial community, and the influences of geochemical properties and the bacterial community as well as metabolism potential have been researched by Tax4Fun2 prediction and high-throughput sequencing.

Point 3: Line 37-38. It´s obviously what? It is obviously that the potential of biochemical remediation of Sb and As contamination in paddy soils profile.

Response 3: Thank you very much for your valuable comments and suggestions. Based on your suggestion, we have carefully checked and modified this sentence. A detailed modification as follows.

Obviously, this study provides a scientific basis for the potential of biochemical remediation of Sb and As contamination in paddy soils profile.

Point 4: Line 45. Biology? Biological process?

Response 4: Thank you very much for your valuable comments and suggestions. Based on your suggestion, we are sure it's a biological process, and have modified it.

Point 5: Line 49. Sb and arsenic (As). I suggest to put Antimony (Sb) and arsenic.

Response 5: Thank you very much for your valuable comments and suggestions. Based on your suggestions, we have modified it.

Point 6: Line 105-108. “Paddy soil was greatly affected by human activities, according to the classification of soil taxonomy, the paddy soil can be divided into either Alfisols or Inceptisols. SL1, SL2 and SL3 were submergenic horizen, percogenic horizen and waterloggogenic horizen, respectively” These are two phrases: 1. Paddy soil was greatly affected by human activities, and 2. The rest…

Response 6: Thank you very much for your valuable comments and suggestions. Based on your suggestion, we have carefully checked and modified the two sentences. A detailed modification as follows.

Paddy soil was greatly affected by human activities, and according to the classification of soil taxonomy, the paddy soil can be divided into either Aqualfs or Aquepts. SL1, SL2 and SL3 were submergenic horizon, percogenic horizon and waterloggogenic horizon, respectively.

Point 7: In this same sentence: horizen or horizon?

Response 7: Thank you very much for your valuable comments and suggestions. Based on your suggestion, we have modified the word.

Point 8: If the soils are Alfisols or Inceptisols, logycally wil be Aqualfs or Aquepts. So I suggest to introduce this soil types at level or suborder.

Response 8: Thank you very much for your valuable comments and suggestions. Based on your suggestion, this part has been modified. A detailed modification as follows.

Paddy soil was greatly affected by human activities, and according to the classification of soil taxonomy, the paddy soil can be divided into either Aqualfs or Aquepts. SL1, SL2 and SL3 were submergenic horizon, percogenic horizon and waterloggogenic horizon, respectively.

Point 9: Line 116: 10ml 1mol·L-1???

Response 9: Thank you very much for your valuable comments and suggestions. Based on your suggestion, we're sure it's like this, which means the volume and concentration of the solution.

Point 10: “Overall, The contamination”, Overall, the contamination

Response 10: Thank you very much for your valuable comments and suggestions. Based on your suggestion, this part has been modified.

Finally, I would like to thank you for your valuable comments and suggestions in the review of the manuscript, and your valuable time in improving the quality of our manuscript.